# Precise tuning of gene expression levels in mammalian cells

Yale S. Michaels [1], Mike B. Barnkob [2], Hector Barbosa[1], Toni A. Baeumler[1], Mary K. Thompson [3], Violaine Andre[2], Huw Colin-York[2], Marco Fritzsche[2,4], Uzi Gileadi[2], Hilary M. Sheppard[5], David J.H.F. Knapp[1], Thomas A. Milne [6], Vincenzo Cerundolo[2] & Tudor A. Fulga[1]

Precise, analogue regulation of gene expression is critical for cellular function in mammals. In contrast, widely employed experimental and therapeutic approaches such as knock-in/out strategies are more suitable for binary control of gene activity. Here we report on a method for precise control of gene expression levels in mammalian cells using engineered microRNA response elements (MREs). First, we measure the efficacy of thousands of synthetic MRE variants under the control of an endogenous microRNA by high-throughput sequencing. Guided by this data, we establish a library of microRNA silencing-mediated fine-tuners (miSFITs) of varying strength that can be employed to precisely control the expression of user-specified genes. We apply this technology to tune the T-cell co-inhibitory receptor PD-1 and to explore how antigen expression influences T-cell activation and tumour growth. Finally, we employ CRISPR/Cas9 mediated homology directed repair to introduce miSFITs into the BRCA1 3′UTR, demonstrating that this versatile tool can be used to tune endogenous genes.

[1] Weatherall Institute of Molecular Medicine, Radcliffe Department of Medicine, University of Oxford, Oxford OX3 9DS, UK. [2] MRC Human Immunology Unit, Weatherall Institute of Molecular Medicine University of Oxford, Oxford OX3 9DS, UK. [3] Department of Biochemistry, University of Oxford, Oxford OX1 3QU, UK. [4] Kennedy Institute of Rheumatology, Nuffield Department of Orthopaedics, Rheumatology and Musculoskeletal Sciences, Oxford OX3 7FY, UK. [5] School of Biological Sciences, University of Auckland, Auckland 1050, New Zealand. [6] Weatherall Institute of Molecular Medicine, MRC Molecular Haematology Unit, NIHR Oxford Biomedical Research Centre Programme, University of Oxford, Oxford OX3 9DS, UK. Correspondence and requests for materials should be addressed to T.A.F. (email: tudor.fulga@imm.ox.ac.uk)

Subtle changes in gene expression can have important biological consequences in mammalian cells[1–3]. However, conventional genetic manipulation strategies such as knockouts and transgenic overexpression are all-or-nothing approaches that fail to recapitulate physiologically relevant changes in gene expression levels. To explore the impact of partial changes in gene expression, fine-tuning systems based on libraries of promoters or ribosome binding sites of varying strengths have previously been constructed in bacteria[4–7] and yeast[4,8]. Here, we set out to develop a tool that would enable precise, stepwise modulation of gene expression levels in mammalian cells. To create a generalisable gene-tuning technology and overcome common limitations of existing genetic manipulation methods we aimed to design a system which: (i) is free from antibiotic triggers, such as doxycycline or rapamycin, which are known to have confounding immunomodulatory effects[9–11] and (ii) does not rely on introducing exogenous siRNAs as these can induce broad off-target effects[12].

To satisfy these design criteria, we sought to harness the exquisite ability of microRNAs (miRNAs) to fine-tune gene expression in mammalian cells. miRNAs are short non-coding RNAs capable of post-transcriptionally controlling gene expression levels by recruiting the RNA induced silencing complex (RISC) to cellular RNAs bearing cognate miRNA response elements (MREs). Importantly, the magnitude of repression depends on the complementarity between a miRNA and its target MRE[13]. We reasoned that by engineering synthetic MREs with varying complementarity to an endogenous miRNA we could precisely modulate expression of user-specified genes without the necessity of supplying any exogenous molecules.

Previous high-throughput screening approaches have enabled in-depth analysis of miRNA expression profiles[14] and the evaluation of contextual features important for miRNA-mediated regulation[15]. Additional studies have described broad functional domains within MREs, such as the "seed" (nt 2–8) and the "supplementary region" (nt 13–16)[13,16]. Because naturally occurring MREs generally bear partial complementarity to their associated miRNAs (and tend to impart only modest regulation over their transcripts) we decided to study how sequence variation in highly complementary synthetic MREs influences the magnitude of miRNA-mediated repression. Similarly to siRNA-mediated silencing, highly complementary MREs are thought to primarily promote cleavage (via Ago2-mediated slicing) or transcript destabilisation[13]. However, it remains unclear how base pairing with each individual nucleotide or pair of nucleotides within such MREs contributes to the degree of gene silencing imparted by a given endogenous miRNA in living cells.

Although MREs with near-perfect complementarity do not commonly occur in mammalian cells, we hypothesise that they could confer strong repression of target transcripts. To enable the forward design of a gene-tuning technology, we develop a high-throughput approach to assess the repressive strength of synthetic MREs at single-base resolution. We identify nucleotides that differentially impact repression and determine that quantifying transcript abundance is sufficient to accurately predict protein output levels. We then use this information to create a panel of miRNA silencing-mediated fine-tuners (miSFITs) and apply them to precisely modulate the expression levels of multiple genes including PD-1, a T-cell co-inhibitory receptor and a target for cancer immunotherapy. We then employ the miSFIT approach to decipher the relationship between antigen levels, T-cell surveillance and tumour growth, an elusive problem in cancer immunology. By fine-tuning a tumour-associated antigen in a mouse melanoma model, we demonstrate that antigen expression level is an important determinant of the anti-tumour immune response in vitro and in vivo. Finally, we use CRISPR/Cas9 to integrate miSFITs into the 3′UTR of the key tumour suppressor gene BRCA1[17], demonstrating that it is possible to achieve genetically-encoded fine tuning of endogenous gene expression levels in mammalian cells.

## Results

**The regulatory landscape of a synthetic miRNA response element.** To develop a fine-tuning system suitable for use in mammalian cells, we sought to redirect endogenous miRNAs to user-defined target mRNAs, thus harnessing the repressive potential of this post-transcriptional regulatory layer. As a proof of concept, we focused on miR-17 which is a well characterised miRNA expressed in numerous human and murine cell types[18,19]. By evaluating the regulatory capacity of a library of synthetic MREs with varying complementarity to miR-17 we reasoned that we could dissect the targeting landscape of this miRNA. The resulting dataset could be used to select MREs of desired strength, providing a generalisable approach for fine-tuning gene expression.

We designed a 23nt degenerate oligonucleotide pool with 91% complementarity to miR-17 and 3% of each alternative nucleotide at every position (Fig. 1a, Supplementary Figure 1). This oligo pool was cloned downstream of a fluorescent reporter (ECFP) in a mammalian expression plasmid and the ensuing MRE variant library was transfected into HEK-293T cells that endogenously express miR-17. We also co-transfected a control reporter bearing an MRE complementary to C. elegans Cel-miR-67, which is not expressed in human cells[20]. After allowing endogenous miR-17 to act on the transcripts templated by the variant library, we harvested mRNA and plasmid DNA (pDNA) and subjected them to targeted deep sequencing (Fig. 1b, Supplementary Figure 1). To estimate the strength of the MRE variants present in our library, we divided their frequency in the mRNA pool by their frequency in the pDNA pool (Supplementary Figure 1).

As expected, MREs with higher complementarity to miR-17 were silenced more effectively (Supplementary Figure 1). Even single nucleotide mismatches diminished silencing by 2.30-fold on average ($+/-$ 0.03, 95% CI) compared to a perfectly matched target (Supplementary Figure 1). To achieve a broad dynamic range in expression levels, we decided to focus our subsequent efforts on single and di-nucleotide MRE variants. First, we analysed all possible single nucleotide variants and asked how each position within the MRE contributes to miRNA-mediated repression (Fig. 1c). As anticipated, certain seed mismatches strongly abrogated silencing, confirming the important role of this region in target selection (Fig. 1c). Intriguingly however, non-seed nucleotides also significantly impacted the degree of repression, with one position even having a greater impact on silencing than most seed nucleotides (Fig. 1c). While the seed region plays a critical role in target selection for native MREs that display low complementarity to miRNAs, our findings suggest that for highly complementary synthetic MREs, non-seed positions may also strongly impact repression. Mutations introducing G:U wobble pairs were always less deleterious to silencing than non-pairing bases, highlighting the importance of thermodynamic stability for miRNA-mediated repression (Fig. 1c). Analysis of double-nucleotide variants revealed that pairs of mismatches within the seed or combinations of seed mismatches with mismatches in positions 14–20 strongly impaired miRNA activity (Fig. 1d). When we subjected a second miRNA (miR-21) to the same high-resolution analysis, the relative importance of each position in the MRE correlated only weakly with miR-17 ($R^2 = 0.22$, $P = 0.03$, linear regression, slope differs from 0, $n = 22$ positions) (Supplementary Figure 2). Notably however, despite this weak correlation, mismatches at

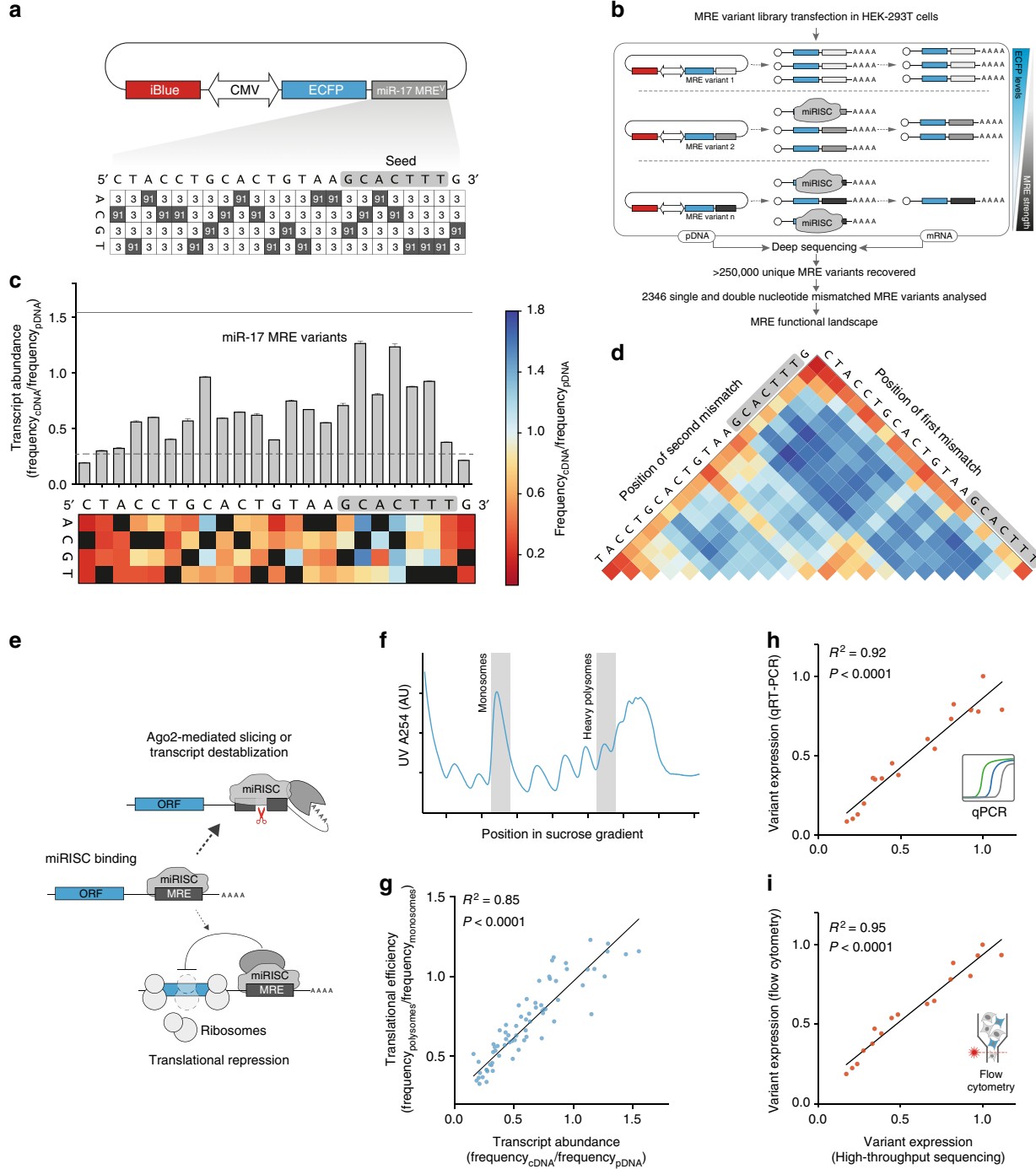

**Fig. 1** Analysis of MRE regulatory landscape at single-nucleotide resolution. **a** MRE reporter library diagram. Values indicate the proportion of nucleotides at each position in the MRE (shaded squares = nucleotides complementary to miR-17). **b** MRE regulatory landscape analysis pipeline. **c** Impact of MRE variants on transcript abundance. Bar graph shows relative contribution of each nucleotide to MRE function, as determined by high-throughput sequencing ($n = 3$ biological replicates, mean + s.d.; dashed line = expression of a perfectly complementary MRE, solid line = expression of a non-targeted MRE- Cel-miR-67). Heat-map displays the effect of each possible mismatch by position and reflects the mean of three replicates (complementary bases are displayed in black). **d** The impact of di-nucleotide substitutions on reporter expression (mean of 3 biological replicates; colour scale is the same as in **c**; grey box = seed region). **e** Schematic representation of the two major pathways underlying miRNA-mediated repression. **f** Polysome profiles generated by sucrose gradient fractionation. Blue trace denotes the spatial distribution of RNA across the gradient as monitored by UV absorbance. Analysed fractions (monosomes and heavy polysomes) are shaded in grey (representative of two biological replicates). **g** Correlation between translational efficiency and transcript stability for all single nucleotide miR-17 MRE variants. $P$ value indicates that the slope of a linear regression model (black diagonal line) significantly differs from 0 ($n = 69$ variants). **h** Linear regression comparing expression measured by high-throughput sequencing with expression measured by RT-qPCR in HEK-293T cells transfected with each of the 15 MRE variants in the validation set ($P < 0.0001$, slope differs from 0, $n = 17$ variants). Expression was calculated by the $\Delta\Delta$CT method using iBlue as a reference gene. **i** Linear regression comparing variant expression measured by high-throughput sequencing and flow cytometry (flow cytometry expression was calculated by normalising ECFP levels to iBlue levels on a single cell basis and taking the mean of that value for each MRE variant) ($P < 0.0001$, slope significantly differs from 0, $n = 17$ variants). Source data are provided as a Source Data file

certain non-seed positions were also able to strongly abrogate silencing by miR-21. Together, these data demonstrate the utility of our high-throughput assay for studying the regulatory strength of highly complementary MREs at single nucleotide resolution. These results also suggest that choosing other input miRNAs for tuning gene-expression may require additional empirical analysis.

This sequencing-based assay allows us to assess the effect of mismatches in synthetic MREs on mRNA stability. In addition to promoting transcript degradation, miRNAs have also been proposed to repress translation[21]. To rule out the possibility that MRE variants were being translationally repressed in a manner that was not predicted by our mRNA/pDNA sequencing approach, we used polysome profiling to isolate monosome-bound and heavy polysome-bound mRNAs[22] (Fig. 1f). We then sequenced cDNA libraries from these fractions and used the ratio of reads in the heavy polysome-bound fraction to reads in the monosome-bound fraction as a measure of translational efficiency for each MRE variant (Fig. 1f, g). This analysis revealed a strong correlation between transcript degradation and translational repression (the inverse of translational efficiency) for single-nucleotide variants in the library (Fig. 1g). This finding suggests that miRNA-target base pairing is a critical determinant of the magnitude of both transcript degradation and translational output for our MRE variant library. However, since miRNAs have been shown to catalyse co-translational target degradation[23], our polysome profiling data cannot directly distinguish between inhibition of ribosome initiation/elongation and degradation of ribosome-bound transcripts. Nonetheless, our results do demonstrate that the mRNA/pDNA sequencing approach is a good predictor of overall synthetic MRE strength.

To further validate the accuracy of our high-throughput MRE screen, we randomly selected 15 single and double nucleotide MRE variants from our library and subjected them to RT-qPCR and flow-cytometry analysis (Fig. 1h, i and Supplementary Figure 3). Both RT-qPCR ($R^2 = 0.92$, linear regression, $n = 17$ variants, Fig. 1h) and flow-cytometry ($R^2 = 0.95$, linear regression, $n = 17$ variants, Fig. 1i) strongly corroborated the high-throughput sequencing analysis, supporting the validity of our screen and confirming the correlation between the strength of transcript degradation and protein output levels in this particular context (Supplementary Figure 3).

**Fine-tuning gene expression levels in human cells**. Next, we sought to demonstrate that our MRE variant library can be used to precisely modulate expression of genes of interest. By ranking all miR-17 MRE variants containing single-nucleotide mismatches, we created a dictionary of microRNA silencing-mediated fine-tuners (miSFITs) that relates MRE sequence identity to ECFP gene expression output in HEK-293T cells (Fig. 2a). Sorting all single-nucleotide miSFIT variants according to their predicted strength revealed that the system has the capacity to achieve precise, stepwise control of gene expression levels (Fig. 2a, Supplementary Figure 4). More specifically, the median difference in expression between adjacent single-nucleotide miSFIT variants is 0.80% of maximal expression (25% percentile = 0.33%, 75% percentile = 1.48%, $n = 69$ variants, 3 biological replicates, Supplementary Figure 4). We repeated this analysis on the 2277 possible di-nucleotide miSFITs and observed that the median difference between adjacent variants was 0.02% (25% percentile = 0.01%, 75% percentile = 0.05%, $n = 2277$ variants, 3 biological replicates, Supplementary Figure 4). This remarkably small step-size between miSFIT variants demonstrates the near-analogue nature of this gene-tuning approach.

We then asked if a selection of miSFIT variants from this dictionary could be deployed to tune expression of proteins other than ECFP. In addition to the 15 randomly selected single and di-nucleotide MRE variants used in previous validation experiments (Supplementary Figure 3) we also included a Cel-miR-67 MRE and 1×, 2×, and 4× perfectly complementary miR-17 MREs. We appended these 19 variants downstream of three independent transgenes in a bi-cistronic expression vector that also encodes a control reporter gene (truncated nerve growth factor receptor, NGFR) that is not under miR-17 control[14]. We chose to tune a second fluorescent protein (EGFP) as well as the T-cell co-inhibitory receptor PD-1 and its cognate ligand PD-L1, two important targets for cancer immunotherapy. The resulting constructs (57 in total) were transfected into HEK-293T cells in triplicate and the expression of each transgene was analysed by flow cytometry (Fig. 2b–d). For all three transgenes, miSFITs conferred stepwise control over expression levels. In addition, the chosen panel provided a broad dynamic range between the highest and lowest expressed construct for each transgene (28-fold, 123-fold, and 28-fold for EGFP, PD-1, and PD-L1, respectively) (Fig. 2b–d). Furthermore, linear regression analysis revealed that the repression exerted by each miSFIT correlated strongly and significantly between each transgene and the original ECFP validation data (Fig. 2e–g).

Next, to demonstrate that miSFITs can stably tune expression levels in another human cell type, we used a selected set of miSFITs to modulate PD-1 expressed from a lentiviral vector in Jurkat T-cells. We transduced a Jurkat cell line that expresses very low levels of PD-1 at baseline with 6 different PD-1-miSFIT constructs as well as a Cel-67 MRE control at low MOI (Supplementary Figure 5). After sorting pools of NGFR[+] (un-repressed internal transduction control) cells, we assayed PD-1 expression by flow cytometry. The selected miSFITs elicited discrete, stepwise control over PD-1 levels (Supplementary Figure 5) in a manner that was predicted by the ECFP MRE dictionary ($R^2 = 0.94$, linear regression, $n = 6$ variants). Together, these data demonstrate that miSFITs are a versatile method for predictably and precisely tuning transgene expression in human cells.

**Modulating tumour antigen expression and T-cell response**. To further illustrate the utility of miSFITs as an effective tool for modulating gene-expression, we next sought to apply this technology towards a biological question that has previously been confounded by technical limitations. More specifically, we set out to explore how peptide-antigen expression levels influence the strength of the anti-tumour immune response in a murine melanoma model. Cancer immunotherapy is a promising class of treatments that aim to enhance anti-tumour cytotoxicity by the adaptive immune system[24]. Sub-types of immunotherapy, including checkpoint blockade and adoptive cell transplant, rely on T-cell receptor (TCR) mediated recognition of peptide antigens presented by MHC-I molecules on the surface of tumour cells[24]. Although in silico algorithms can accurately predict which peptide antigens are likely to elicit an immune response[25], understanding how peptide-antigen expression levels influence the strength of the antitumour immune response in vivo remains elusive. A quantitative analysis of this relationship could provide an important benchmark for predicting which tumours might respond to anti-cancer immunotherapy.

Previous efforts to titrate peptide-MHC concentrations have relied on coating culture vessels with recombinant peptide-MHC multimers[26] or on briefly adding varying concentrations of peptide to cellular growth media (a process known as peptide pulsing)[27]. Although valuable, these methods cannot accurately

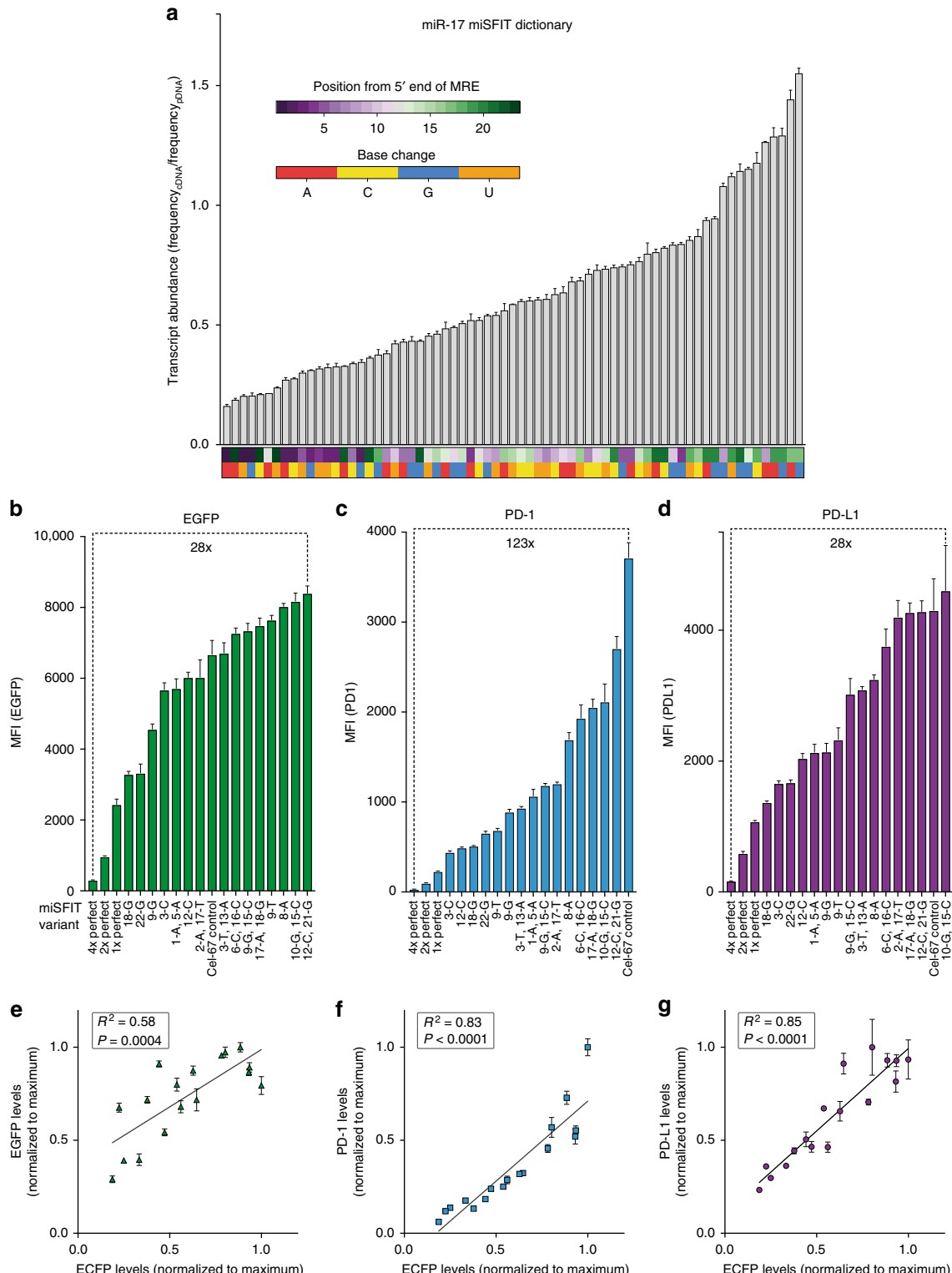

**Fig. 2** Synthetic miSFIT variants enable fine-tuning of gene expression in human cells. **a** Impact on transcript abundance of all single-nucleotide miR-17 miSFIT variants ranked by expression output. Coloured rectangles beneath each bar indicate the position (top) and base change (bottom) of the synthetic MRE variant (n = 3 biological replicates, mean + s.d.). **b–d** Flow cytometry analysis of HEK-293T cells transfected with a panel of 19 miR-17 miSFIT variants placed in the 3′UTR of EGFP (**b**), PD-1 (**c**), and PD-L1 (**d**) (n = 3 biological replicates, mean + s.d.). Fold-change between maximum and minimum expression is noted for each transgene. **e–g** Linear regression analysis of miSFIT strength correlation between ECFP-EGFP (**e**), - PD-1 (**f**), and - PD-L1 (**g**) in HEK-293T cells (n = 3 biological replicates, mean + s.d.). P values indicate slopes significantly differ from 0. Source data are provided as a Source Data file

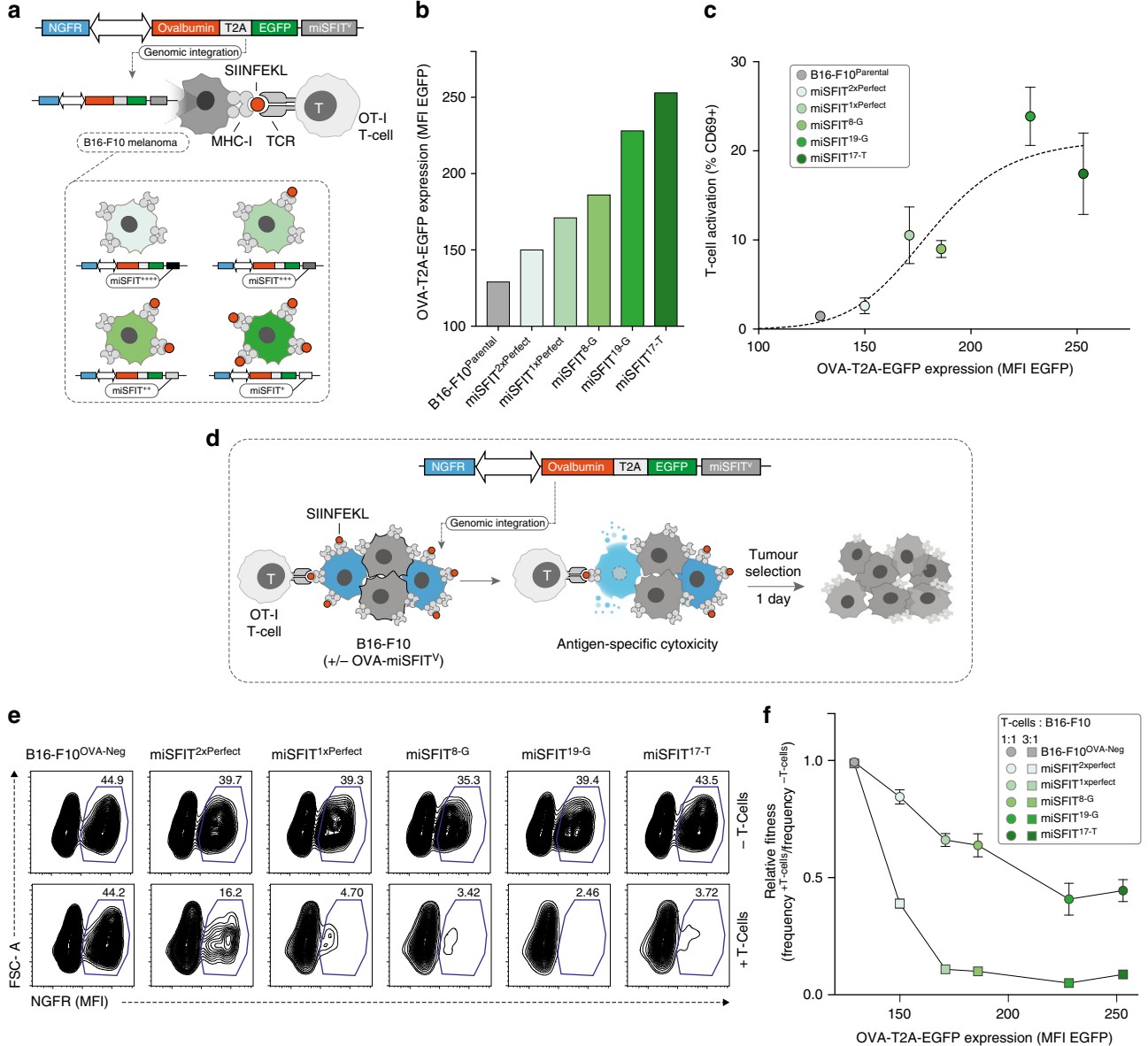

**Fig. 3** Fine-tuning antigen expression levels and T-cell activity. **a** Strategy for tuning Ovalbumin (OVA) expression in B16-F10 melanoma cells using lentivirally integrated miSFITs. Red circles represent SIINFEKL, a peptide antigen derived from ovalbumin. OT-I T-cells express a TCR specific for SIINFEKL presented on MHC-I. **b** Flow cytometry analysis of OVA-T2A-EGFP expression in B16-F10 cell lines transduced with six different miSFIT variant (miSFIT$^V$) lentiviruses (see Supplementary Figure 6 for the gating strategy and distribution of fluorescence intensity) **c** CD8$^+$, OT-I T-cell activation by OVA-miSFIT B16-F10 cell lines. CD69 expression was quantified by flow-cytometry ($n = 5$ biological replicates, mean $+/-$ s.d.). **d** Schematic representation of mixed-culture experimental design. OVA-negative (NGFR$^-$) are mixed with OVA-miSFIT (NGFR$^+$) B16-F10 cells and are challenged overnight with OT-I T-cells. **e** Representative flow cytometry plots of mixed culture experiments. The percentage of NGFR$^+$ (OVA-miSFIT) cells (blue polygon gate) surviving after overnight selection in the presence or absence of CD8$^+$, OT-I T-cells(at a ratio of 3:1 T-cells to B16-F10 cells) is indicated for each condition. **f** Relative fitness of B16-F10 cell lines as a function of OVA expression. Relative fitness was calculated by dividing the frequency of NGFR$^+$ cells with T-cells by the frequency of NGFR$^+$ cells without T-cells ($n = 3$ biological replicates, mean $+/-$ s.d.). Source data are provided as a Source Data file

re-capitulate the endogenous pathway of antigen expression, proteolytic processing and subsequent surface presentation. Furthermore, because peptide pulsing is inherently transient, this method precludes tracking the survival of antigen-expressing cells in vivo. To understand how antigen-expression influences the anti-tumour immune response and the relative fitness of cancer cells in vitro and in vivo, we used miSFITs to finely tune expression of ovalbumin (OVA), a model immunogenic protein, in a stable and physiologically accurate fashion.

To this end, we created a panel of seven bi-cistronic OVA expression vectors, each encoding a distinct miSFIT variant in the

3′UTR of ovalbumin (Fig. 3a). We also coupled EGFP downstream of ovalbumin via a self-cleaving T2A peptide, enabling us to monitor expression levels by flow-cytometry (Fig. 3a). In each vector, NGFR was included as an unsilenced internal control reporter. We transiently expressed these constructs in B16-F10 melanoma cells to evaluate gene expression output (Supplementary Figure 6). This analysis revealed discrete, stepwise tuning of target levels, although the exact ranking of miSFIT variant strength differed from what we observed when tuning PD-1 in human Jurkat T-cells (Supplementary Figure 6). To generate stable cell lines expressing varying levels of ovalbumin we then

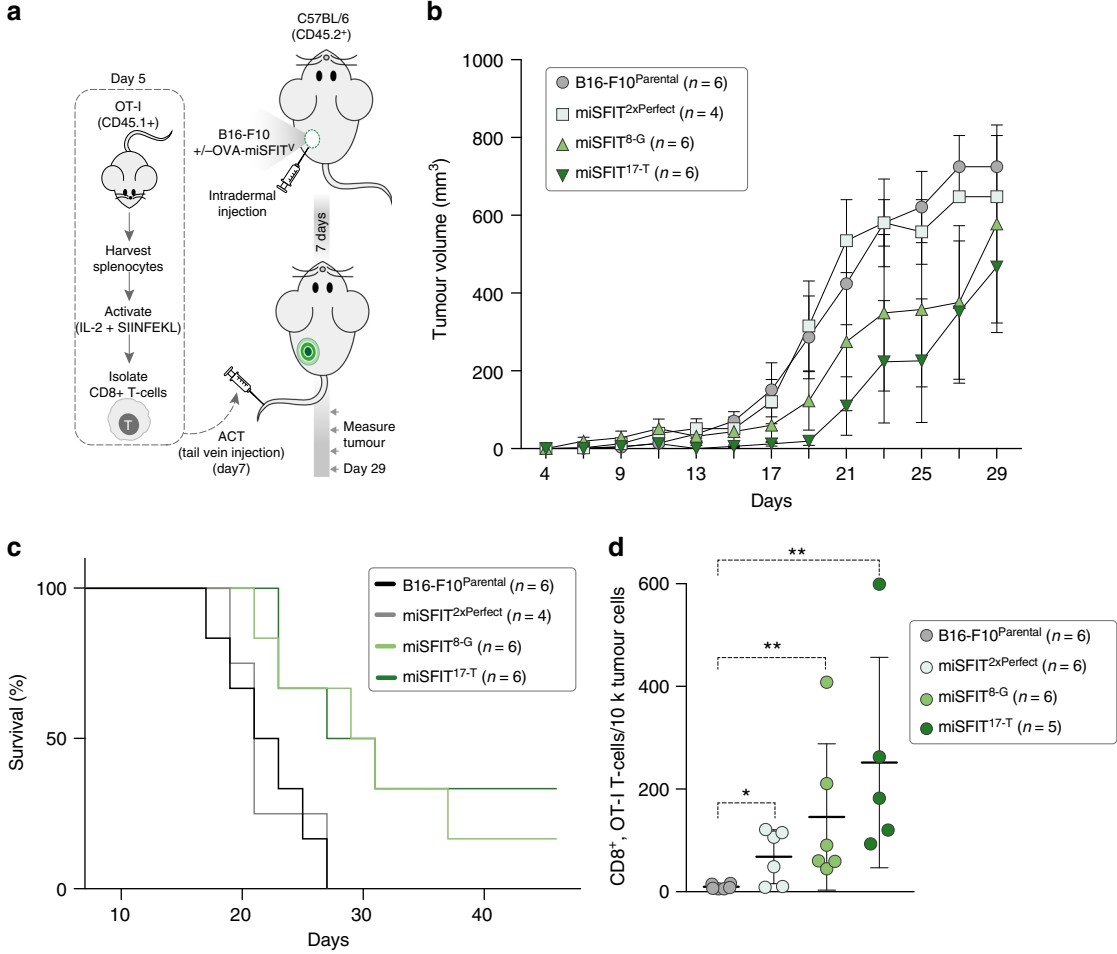

**Fig. 4** Antigen expression levels determine the anti-tumour immune response in vivo. **a** Experimental design for in vivo OVA-miSFIT B16-F10 tumour growth experiments. **b** Analysis of tumour volume over time for four B16-F10 cell lines (3 OVA-miSFIT variant lines and one B16-F10 parental control line) challenged with OT-I CD8$^+$ T-cells (x-axis = number of days from tumour cells injection; mean +/− s.e.m.) **c** Survival curves for mice injected with B16-F10 lines following the same experimental setup as in **a**. **d** Frequency of CD8$^+$, OT-I TILs per tumour (mean +/− s.d., Mann–Whitney U-test, *P < 0.05, **P < 0.01). For experimental setup see Supplementary Figure 8. Source data are provided as a Source Data file

transduced B16-F10 cells with a subset of five OVA-miSFIT constructs at low MOI (<2% transduction efficiency). The semi-random nature of lentiviral integration[28] results in heterogeneity of gene expression between individual cells. To mitigate this effect, we sorted and expanded pools of 150,000 cells on the basis of NGFR expression. After confirming that we successfully tuned ovalbumin expression in the resulting five cell lines (Fig. 3b), we asked how antigen expression levels influence CD8$^+$ T-cell activation.

The OT-I T-cell receptor (OT-I) is specific for SIINFEKL, a short peptide antigen derived from ovalbumin, presented by MHC-I[29]. We co-cultured each of the five B16-F10 lines expressing differential ovalbumin levels and the OVA-negative parent line with CD8$^+$ OT-I T-cells and assayed activation by measuring CD69 expression (Fig. 3c). Indeed, increasing OVA expression resulted in a concomitant increase in the proportion of activated T-cells, presumably due to the greater probability of each T-cell encountering and responding to a SIINFEKL-MHC-I complex (Fig. 3c).

Under selective pressure by the adaptive immune system, tumours have been shown to acquire mutations that prevent effective T-cell surveillance in a process known as immunoediting. This is generally achieved through loss-of-function mutations in MHC genes, upregulation of immunosuppressive molecules, or by elimination of clones expressing neo-antigens[30]. In addition to

these reported phenomena, we hypothesised that tumour cells might also be selected on the basis of antigen expression levels. To address this question, we first mixed the five OVA-miSFIT B16-F10 cell lines at a 1:1 ratio with OVA-negative B16-F10 cells (Fig. 3d). We then allowed these mixed cultures to grow overnight in the presence or absence of OT-I T-cells. Because all OVA-miSFIT lines express NGFR whilst the OVA-negative parent line does not, we quantified the relative abundance of OVA$^+$ (NGFR$^+$) and OVA-negative (NGFR$^-$) cells following the T-cell challenge (Fig. 3e, Supplementary Figure 6). Tuning antigen expression using miSFITs modulated the strength of T-cell mediated selection in a dose-responsive manner at two T-cell: tumour cell ratios (Fig. 3e, f). Notably, even low antigen expression was sufficient to elicit a strong reduction in relative fitness at a high T-cell: tumour cell ratio (Fig. 3f). Stably integrating miSFITs into the genome by lentiviral transduction did not result in substantial competition for endogenous miR-17, affirming that observed differences in fitness were due to differential OVA expression levels (Supplementary Figure 7).

**Antigen expression level dictates tumour growth and survival.**
Next, we asked if the effect of antigen expression on melanoma survival that we observed in vitro correlates with tumour growth rates in vivo. First, we injected a subset of our engineered OVA-

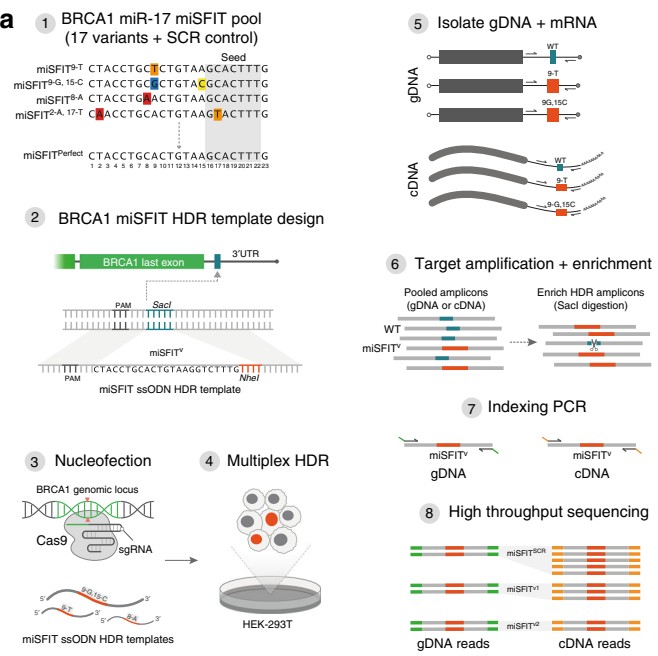

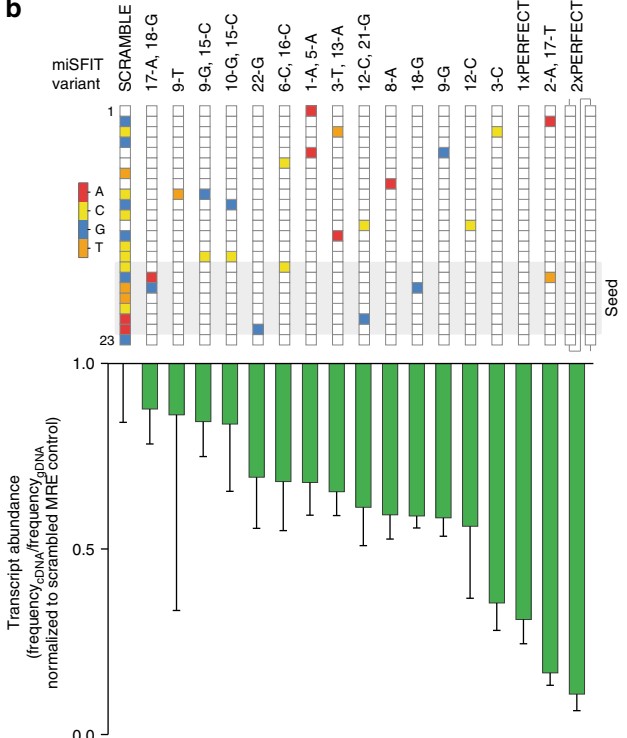

**Fig. 5** CRISPR/Cas9-mediated miSFIT knockin enables fine-tuning of endogenous BRCA-1 expression. **a** Experimental workflow for tuning endogenous gene expression by genomic integration of miSFIT regulatory elements in the native 3′UTR of a target gene using CRISPR-mediated HDR. Assessing the impact of miSFIT variants is carried out in a pooled format by high throughout sequencing. **b** Impact of a panel of 17 miR-17 miSFIT variants on endogenous BRCA-1 expression. ($n = 3$ biological replicates, mean—s.d.). The position and identity of mismatched bases in each variant is indicated by the colour code. Bar plot reflects the degree of BRCA-1 repression imparted by each miSFIT variant relative to the scrambled MRE control. Source data are provided as a Source Data file

miSFIT-B16-F10 cell lines into syngeneic recipient mice (Fig. 4a). After allowing intradermal tumours to establish for 7 days, we adoptively transferred CD-8[+] OT-I T-cells and monitored tumour growth for an additional 22 days (Fig. 4a). Antigen expression levels impacted tumour growth in vivo in a manner that faithfully mirrored in vitro T-cell activation and killing (Fig. 4b). Due to the intrinsic variability in growth of the B16-F10 tumour in vivo, our experiments did not have the statistical power to detect significant differences in tumour volume between pairs of miSFIT cell lines ($P > 0.05$, two-way Anova of tumour volumes at day 19, $n \geq 4$ tumours). However, linear regression analysis revealed that tumour volume did vary significantly with OVA levels across all B16-F10 miSFIT lines tested ($P = 0.01$, slope significantly differs from 0, $n \geq 4$ tumours).

We continued to monitor mice for 46 days and observed that antigen expression markedly influenced survival ($P = 0.0038$, Logrank test for trend, $n \geq 4$ mice, Fig. 4c). Mice bearing tumours with no, or low antigen expression all met our endpoint criteria by day 27. In contrast, medium or high OVA expressing tumours displayed a substantial increase in survival. One third of the mice bearing high-antigen B16-F10-OVA cells (2/6) survived for 46 days with tumours that were nearly undetectable by the experiment's endpoint (Fig. 4c). Together, these findings illustrate the importance of tumour-associated antigen expression levels in determining the strength of the immune-response.

To understand why higher antigen expressing tumours were more effectively controlled, we harvested and analysed tumour infiltrating lymphocytes (TILs) at eight days after adoptive T-cell injections (Supplementary Figure 8). Changes in antigen expression levels differentially affected expression of CD69, CD25, PD-1, and CTLA-4 on OT-I T-cells (Supplementary Figure 8). Importantly, increasing levels of OVA lead to a dose-responsive increase in the frequency of TILs in vivo ($P = 0.0017$, linear regression, slope significantly differs from 0, $n \geq 5$ tumours) (Fig. 4d, Supplementary Figure 8). These findings highlight the value of miSFIT technology in studying intracellular interactions and cellular fitness, and demonstrate the role of tumour-associated antigen levels in controlling T-cell infiltration, tumour growth and survival.

**Tuning endogenous BRCA1 expression with miSFITs**. Previously reported methods for modulating gene expression levels in mammalian cells such as drug-inducible promotors[31] and artificial upstream open reading frames (uORFs)[32] have been successfully applied towards tuning transgenes. However, technologies for tuning endogenous gene expression are currently lacking. Owing to their short sequence length and to the fact that miSFITs can be introduced at flexible locations within endogenous 3′UTRs, these elements are ideally suited for genomic integration by CRISPR-mediated homology directed repair (HDR). To demonstrate this possibility, and to expand the scope of this technology to endogenous gene regulation, we used CRISPR/Cas9 to integrate a panel of miR-17 miSFITs into the native 3′UTR of BRCA1 (Fig. 5a). BRCA1 is a key tumour suppressor that plays an important role in DNA repair and reduced BRCA1 expression is associated with breast and ovarian cancer[17,33].

To tune endogenous BRCA1 expression levels without altering the protein coding sequence, we designed single stranded oligonucleotide (ssODN) HDR donors containing one of 18 miSFIT inserts. In addition to a set of 15 previously selected single and di-nucleotide miSFIT variants, we also designed HDR donors comprising 1× or 2× perfectly complementary miR-17 MREs and a scrambled control. These 18 ssODNs were pooled and co-delivered to HEK-293T cells together with a plasmid that expresses Cas9 and a sgRNA targeting the BRCA1 3′UTR.

Following a 72-h incubation, we isolated genomic DNA (gDNA) and mRNA from three nucleofection replicates and PCR amplified the target locus. Because the ssODNs were designed to ablate an endogenous SacI recognition site, we were able to enrich for edited gDNA and cDNA by restriction digestion (Fig. 5a). The undigested PCR products were purified, indexed and analysed by HTS.

To quantify the impact of the selected miSFITs on BRCA1 expression we normalised the abundance of each variant in the cDNA libraries to their respective abundance in the gDNA libraries[34,35]. We compared the expression level of each miR-17 miSFIT to that of a scrambled control. Our panel of miSFIT variants reproducibly conferred discrete, stepwise downregulation of BRCA1 expression (Fig. 5b). The weakest miSFIT reduced BRCA1 levels by 12% (SD = 9%) while the strongest variant reduced expression by 89% (SD = 5%). These data demonstrate that miSFITs are an effective tool for tuning expression of endogenous genes providing a similar level of control to that observed for transgene expression.

## Discussion

Here we have developed a powerful tool for tuning gene expression output in mammalian cells and used it to uncover a critical role of cancer antigen expression in modulating the immune response. It should be noted that the ovalbumin-derived model antigen SIINFEKL is recognised by the OT-I TCR with very high affinity. However, patient derived tumour-associated antigens have varying affinity and avidity for their cognate TCRs. Applying the miSFIT technology to bona-fide tumour antigens will enable scientists to understand how antigen immunogenicity[36] and expression levels interact to influence the immune response. In turn, such studies could allow clinicians to better predict how tumours will respond to immunotherapy.

Although miSFITs enabled precise control of all genes that we tested, in the case of OVA expression in B16-F10 cells, some of our MREs elicited stronger or weaker silencing than expected based on the synthetic MRE dictionary (compare Fig. 3b to Supplementary Figure 5). Differences in miSFIT strength between organisms may be the result of differences in RISC composition, miR-17 family member expression or the distinct repertoire of endogenous RNA binding proteins between human (HEK 293T, Jurkat T-cells) and murine (B16-F10) cells. To demonstrate that it is still possible to robustly tune expression levels even when miSFITs do not behave as predicted, we tested a panel of 18 PD-1-miSFIT variants in B16-F10 cells. This panel conferred precise, stepwise tuning of PD-1, despite the fact that the relative order of repression did not correlate with what we observed for the same miSFIT variants in HEK-293T cells (Supplementary Figure 9).

Customised applications of this technology may need to be validated and modified in some instances. However, we propose that the panel of miR-17 miSFITs used in this study will be suitable for a broad range of applications. miR-17 is expressed in all tissues and cell lines for which data are available in miRmine, a public collection of miRNA expression studies[19]. In addition, since miSFITs respond to endogenous miRNAs, this technology may also enable contextual tuning of target genes in specific cell populations[37] or in response to defined physiological stimuli. This could be accomplished by simply designing miSFITs complementary to miRNAs that display cell-type/cell-state specific expression patterns.

We anticipate that in complementing existing methods such as uORFs, inducible promotors and siRNAs, miSFIT technology will advance the scope and versatility of gene tuning in mammalian cells. Notably, this platform also displays a number of advantages over existing methods for manipulating expression levels. Unlike titratable promotors, miSFITs do not require chemical inducers that have confounding effects on cellular metabolism and are difficult to dose in vivo[9]. Controlling viral multiplicity of infection might allow coarse control over gene expression. However, as reported in this study, even at single copy integration, strong viral promotors instigate over-expression above physiologically relevant levels. Unlike exogenously delivered siRNAs, miSFITs co-opt endogenous miRNAs to regulate gene expression. Since the sequence space of endogenous miRNAs is several orders of magnitude smaller than that of the transcribed genome, miSFITs are not confounded by the off-target specificity issues associated with siRNAs.

Notably, the short length of miSFIT elements makes them amenable to integration into genomic loci using CRISPR/Cas9 HDR with ssODNs. This property allowed us to tune native BRCA1 expression in a precise, stepwise fashion, demonstrating the ability to control the levels of an endogenous gene by a genetically encoded synthetic system. This represents a unique capability that has not been demonstrated using previous technologies.

In addition to their value as a research tool, miSFITs may have future therapeutic applications. For example, gene-expression levels can influence the efficacy of anti-cancer immunotherapy. On tumour-reactive T-cells, high PD-1 expression suppresses effector function[38]. Therapies that block PD-1 signalling can improve the anti-tumour immune response[38] but also instigate adverse autoimmunity events in a large proportion of patients[39]. Using miSFITs to fine-tune endogenous PD-1 levels in patient-derived effector T-cells might achieve an optimum balance between exhaustion and autoimmunity, enabling safer and more effective adoptive cell therapy. Similarly, miSFITs could be applied to other co-inhibitory receptors like CTLA-4 or to therapeutic transgenes such as Chimeric Antigen Receptors (CARs). A recent report suggests that CAR efficacy depends on achieving an optimised expression level that prevents tonic signalling and exhaustion[2]. miSFITs offer a potential method for simply and precisely controlling CAR expression in patient-derived T-cells. Because of their versatility and ease of implementation, miSFITs hold promise for tuning expression of a wide range of genes with applications in basic research and therapeutic cellular engineering.

## Methods

**MRE variant library construction**. The sequences of all oligonucleotides used in this study are listed in Supplementary Dataset 1. We purchased hand-mixed, partially degenerate oligonucleotides from Integrated DNA Technologies (IDT) comprising a constant flanking region and a variable region with partial complementarity to either hsa-miR-17 or hsa-miR-21. In this study, the term synthetic MRE refers to a sequence of equal length, and largely complementary to a given miRNA. Within this library, there are 69 possible single nucleotide variants for a 23nt sequence (at each position, there are three possible ways to introduce a mismatch). For a sequence length $n$, the number of combinations with r mismatches is given by the formula:

$$C(n, r) = \frac{n!}{(r!(n-r)!)} \quad (1)$$

Therefore, for a 23nt sequence there are 253 combinations of di-nucleotide mismatches. Each base can be changed in 3 possible ways giving 2277 total combinations of mismatches ($253 \times 3 \times 3$). To achieve maximum coverage of single and double nucleotide variants, 91% of the complementary base and 3% of each non-complementary base were incorporate at each position in the MRE libraries. Each degenerate oligo was PCR amplified in triplicate using Phusion High-Fidelity PCR Master Mix with GC Buffer (NEB), using primers miR17_Lib_Gen_F and miR17_Lib_Gen_R, which append BsmBI recognition sites on both sides of the MRE. The resulting PCR products were pooled and purified using the MinElute PCR Purification Kit (Qiagen).

We performed a large-scale restriction cloning reaction to ligate the degenerate MRE PCR product into a reporter plasmid. The reporter plasmid comprises a bidirectional CMV promotor driving expression of iBlue fused to a degradation

signal derived from Ornithine Decarboxylase and ECFP fused to the same degradation signal. We linearised 10.5 µg of the reporter plasmid downstream of ECFP by digesting with BsmBI.

The degenerate MRE PCR product (300 ng) was cut with BsmBI and ligated to the linearised, dephosphorylated (Antarctic Phosphatase, NEB) and gel purified (QIAquick Gel Extraction Kit, Qiagen) reporter plasmid using T4 DNA Ligase (NEB) at 16 °C overnight. We purified the ligations using the QIAquick PCR Purification Kit (Qiagen) and transformed approximately 3.6 µg of the purified product into 10-beta Electrocompetent E.coli (NEB) following the manufacturer's instructions. Transformants were plated overnight at 32 °C on 24.5 cm$^2$ ampicillin-treated LB agar plates. We recovered the resulting plasmid library using the QIAfilter Plasmid Midi Kit (Qiagen).

**HEK-293T cell culture and MRE variant library transfection**. HEK-293T cells (purchased from ATCC, ATCC-CRL-11268) were grown in Dulbecco's modified Eagle's medium (DMEM, Gibco) supplemented with 15% FBS (GIBCO) and 1% Penicillin-Streptomycin (P/S, 10,000 U/mL, Gibco). We screened cells for mycoplasma at the outset of the project. Cells were seeded in 12-well plates, 24 h prior to transfection, allowing them to reach 80–90% confluency on transfection day. On the day of transfection, we replaced complete growth media with DMEM, 2% FBS (no P/S). We prepared three independent transfection mixtures, each containing 4 µg of the degenerate MRE reporter library, 4 ng of miR-Cel-67 MRE control plasmid and 12 µL polyethylenimine (PEI, 1 mg/mL, Sigma-Aldrich) in 400 µL Opti-MEM (Gibco). Each mixture was applied dropwise to 4 wells of a 12-well plate and incubated for 24 h.

**Polysome profiling**. To generate enough cell lysate for polysome profiling, we seeded HEK-293T cells in two independent 15 cm$^2$ culture dishes, allowing them to reach 70–80% confluency by the day of transfection. For each dish, we combined 25 µL each Lipofectamine 3000 and P3000 Reagent (Thermo Fisher) with 12.5 µg of the degenerate MRE reporter library and 100 ng of miR-Cel-67 MRE control plasmid, transfected according to the manufacturer's instructions and incubated for 24 h. To arrest translation, cycloheximide (CHX, Merck) was added to the culture dishes at 100 µg/mL for 10 min at 37 °C. Next, dishes were placed on ice and washed with cold PBS (Life Technologies) supplemented with CHX (100 µg/mL). We scraped the dishes in PBS + CHX (100 µg/mL), centrifuged the harvested cells at 1000 × g for 3 min at 4 °C and discarded the supernatant. Cell pellets were re-suspended in 200 µL of hypotonic lysis buffer (10 mM HEPES pH 7.8, 1.5 mM MgCl$_2$, 10 mM KCl, 0.5 mM DTT, 1% Triton X-100 and 100 mg/mL CHX) and incubated for 5 min on ice. Next, we lysed the cells with 10 strokes through a 26 gauge needle and pelleted the nuclei by centrifuging at 1500×g for 5 min at 4 °C. The supernatant was flash frozen in liquid nitrogen and stored at −80 °C.

10–50% (W/V) sucrose gradients were generated using a Gradient Master (Biocomp Instruments) from 10% and 50% sucrose solutions in gradient buffer (100 mM KCl, 5 mM MgCl$_2$, 20 mM HEPES-KOH pH7.5, 1 mM DTT, 100 µg/mL CHX). We thawed the cell lysates and layered them on top of the chilled sucrose gradients before centrifuging at 4 °C for 2 h at 36,000 RPM (160,030 × g average) in a SW-41 rotor. Gradients were fractionated from the top using a Gradient Fractionator (Biocomp Instruments). To recover RNA from the resulting fractions we added 2.25 volumes of 8 M Guanidine HCl (Sigma-Aldrich) and vortexed the samples. Next, 3.25 volumes of isopropanol were added and samples were incubated overnight at −20 °C. Reactions were centrifuged at >13,000 × g for 30 min at 4 °C and the supernatant was aspirated. RNA pellets were re-suspended in a mixture of 90 µL nuclease free H$_2$O (Invitrogen), 10 µL 3 M Sodium Acetate (Invitrogen) and 1 µL 5 mg/mL glycogen (Ambion) and precipitated with 250 µL of cold 100% ethanol (VWR). After 30 min incubation on ice, samples were centrifuged at >13,000 × g for 30 min at 4 °C. Next, pellets were washed with 500 µL of 70% ethanol and re-suspended in nuclease free H$_2$O.

**pDNA and cDNA library prep and high-throughput sequencing**. We used the All Prep DNA/RNA Mini kit (Qiagen) to simultaneously extract plasmid DNA (pDNA) and mRNA from HEK-293T cells transfected with the degenerate MRE reporter library. After performing a genomic DNA wipe-out, cDNA was generated from mRNA and polysome-associated RNA using the QuantiTect Reverse Transcription kit (Qiagen) following the manufacturer's instructions. To create amplicon libraries for high-throughput sequencing, the degenerate MRE and a short flanking region were PCR amplified using the primers bi-dir-Miseq-F and bi-dir-Miseq-R. For cDNA and pDNA we used Phusion High-Fidelity PCR Master Mix with GC Buffer (NEB) and the following cycling conditions: initial denaturation (98 °C for 30 s), 23 amplification cycles (98 °C for 10 s, 65 °C for 10 s, 72 °C for 10 s) and final extension (72 °C for 5 min). We used 20 ng pDNA and cDNA generated from 200 ng of RNA as input for these PCRs. For cDNA from RNA recovered from polysome fractions we used KAPA HiFi HotStart ReadyMix (Fisher Scientific) and the following cycling conditions: initial denaturation (98 °C for 30 s), 21 amplification cycles (98 °C for 10 s, 65 °C for 10 s, 72 °C for 10 s) and final extension (72 °C for 5 min). cDNA from 100 ng of polysome-associated RNA was used as input for each PCR. These initial PCR products were gel-purified using the QIAquick Gel Extraction Kit (Qiagen). We diluted the recovered product between 10 and 30 fold depending on band intensity.

To make amplicon libraries compatible with Illumina machines, we performed a second PCR to append TruSeq index sequences and p5/p7 adaptors to each amplicon. We used a dual barcoding strategy where a unique combination of forward and reverse index primers were assigned to each biological sample. We performed the PCRs with Phusion High-Fidelity PCR Master Mix with GC Buffer (NEB) and the following cycling conditions: initial denaturation (98 °C for 30 s), 13 amplification cycles (98 °C for 10 s, 62 °C for 10 s, 72 °C for 10 s) and final extension (72 °C for 5 min).

The manufacturer reports an error rate of $9.5 \times 10^{-7}$ for Phusion High-Fidelity PCR Master Mix in GC Buffer. After 36 total PCR cycles, we expect an aggregate error rate of $<3.6 \times 10^{-5}$. This error rate will not impact our estimates of MRE strength in any meaningful way. Importantly, the high reproducibility between library replicates and the strong correlation between HTS data and RT-qPCR validation experiments both support the notion that our findings are not encumbered by PCR error or amplification bias.

We used Agencourt AMPure XP beads (Beckman Coulter) at 0.75× to purify the amplicon libraries which we subsequently quantified using the Qubit dsDNA HS Assay Kit (ThermoFisher Scientific). The samples were sequenced (150 bp PE sequencing) on either the HiSeq4000 (Illumina) or the MiSeq v2 (Illumina).

**High-throughput sequencing data analysis**. High-throughput sequencing data were analysed using R (Version 3.4.1) and all scripts are available upon request. After inspecting the quality of sequencing data with FastQC, we used the Biostrings package (version 2.44.2) to trim reads down to the MRE and subsequently count the occurrence of each type of variant of interest in all amplicon libraries. At least 6000 reads were recovered for all 69-possible single-nucleotide variants in each pDNA library. At least 100 reads were recovered for all 2277 possible di-nucleotide MRE variants in each pDNA library (Mean = 315.56, SD = 85.74). We calculated variant frequency by normalising read counts of each variant of interest to total library read counts in the respective library. We calculated transcript abundance for each variant by dividing its read frequency in the cDNA library to its read frequency in the respective pDNA library. We calculated translation efficiency for variants present in polysome profiles by dividing their read frequency in the heavy-polysome-bound library by read frequency in the respective monosome-bound library.

**Validation of high-throughput sequencing results by RT-qPCR**. To validate our high-throughput sequencing assay we randomly selected miR-17-MRE variants by screening colonies from a 1/30,000 dilution of the variant library by Sanger sequencing using primer bi-dir-MRE-seq-1. Colonies were screened until we identified 15 unique single and double nucleotide variants. The randomly selected variants reflect an unbiased representation of the total variant library. Of note, the average read count in pDNA libraries for the double nucleotide variants in this random validation set is 323.87 (SD = 81.8) which is consistent with the mean read count for all possible double nucleotides.

These constructs were individually transfected into HEK-293-T cells in triplicate in addition to control reporters encoding a Cel-miR-67-MRE and a perfectly complementary miR-17 MRE using the PEI transfection method described above. Twenty-four hours after transfection we extracted RNA using the RNeasy Mini Kit (Qiagen). For each replicate, cDNA was generated from 100 ng of total RNA using the QuantiTect Reverse Transcription kit (Qiagen). We performed RT-qPCR using the SsoAdvanced Universal SYBR Green Supermix kit (Bio-Rad) on a CFX384 real-time system (Bio-Rad) with primer pairs spanning the MRE (MRE_qPCR-F and MRE_qPCR-F) or within the iBlue transcript (iBlue_qPCR-F and iBlue_qPCR-F) which serves as an internal control. The ΔΔCt method was used to compare expression of all MRE variants to a Cel-67-MRE control reporter by comparing the Ct of ECFP to that of iBlue for each sample replicate.

**Lentiviral vector cloning and virus production**. We generated EGFP, PD-1, and PD-L1 lentiviral expression vectors using standard restriction cloning methods. The parent vector AB.pCCL.sin.cPPT.GFP.miR-17-3p.sensor.PGK.dNGFR.WPRE was a gift from Brian Brown (Addgene plasmid #85866). To simplify subsequent cloning steps a SbfI recognition site was introduced downstream of the minimal CMV promotor. The human PD-1 and PD-L1 ORFs were amplified from the PD-1 and PD-L1 BRET vectors respectively (a generous gift from Simon Davis) using the primers PD1_Lenti_Shuttle_F/PD-L1_Lenti_Shuttle_F and PDL1_Lenti_Shuttle_R. We digested these PCR products, as well as the destination vector with SbfI and NheI and ligated them using T4 DNA ligase (NEB).

The In-Fusion HD Cloning System (Takara Clontech) was used to replace PD-1 in the lentiviral expression vector with cytoplasmic-localised ovalbumin (OVA) coupled to EGFP by a T2A peptide cleavage signal to create a OVA-T2A-EGFP vector. We PCR amplified T2A-EGFP from pX458 (A gift from Feng Zhang, Addgene plasmid #48138) with the primers GFP_in_fusion_F2 and EGFP-in-fusion-R. OVA (without the first 47 amino acids) was amplified from the OVACyt vector[40] using the primers Ova_In_Fusion_R2 and Ova-in-fusion-F. We fused the parent vector (linearised with SbfI and NheI) with the two inserts following the In-Fusion manufacturer's instructions. To create miSFIT-tuning vectors we generated MRE inserts from short oligonucleotides (IDT). MRE inserts were annealed and phosphorylated (T4 PNK, NEB) and introduced downstream of PD1 or OVA-

T2A-EGFP by restriction cloning between NheI and AgeI. MRE insertion was confirmed by Sanger sequencing using primer BBBdir-seq-2.

To produce lentiviral particles in HEK-293T cells, we co-transfected each lentiviral transfer vector with pCMV-dR8.91 and pMD2.G at a ratio of 1.5:1:1 using Polyethylenimine (PEI, 1 mg/mL, Sigma-Aldrich) as described above. After 24 h we exchanged the transfection media (DMEM, 2% FBS, no P/S) with full media (DMEM, 15% FBS). We collected and filtered (0.22 μm filter, Millipore) viral supernatant 24 h later and stored it at −80 °C until transduction. We transduced B16-F10 cells and Jurkat T-cells using un-concentrated viral supernatant. Jurkat T-cells (clone 1.G4, a gift from Simon Davis) were maintained in RPMI-1640 media (Gibco) supplemented with 10 mM HEPES (Life Technologies), 1 mM Sodium Pyruvate (Life Technologies), and 15% Foetal Bovine Serum (FBS, GIBCO). B16-F10 melanoma cells (ATCC-CRL-6475) were grown in DMEM (Gibco) supplemented with 15% FBS (Gibco). Jurkat and B16-F10 cells were screened for mycoplasma at the outset of the project.

**Flow cytometry and fluorescence-activated cell sorting**. All flow-cytometry experiments were performed on the BD LSR Fortessa Analyzer or the FAC-Symphony (BD Biosciences) and data were analysed using FlowJo (Version 10.3.0). We harvested adherent cells (B16-F10 or HEK 293T) using 0.05% Trypsin with EDTA (Thermo Fisher Scientific). For experiments requiring antibody staining we washed cells with FACS buffer (PBS with 5% FBS) before and after staining. For an overview of our flow-cytometry gating strategies see Supplementary Figures. To generate miSFIT cell lines we transduced cells at low multiplicity of infection (For Jurkat T-cells < 15% transduced, for B16-F10s < 3% transduced), waited 5–7 days and selected stably transduced cells by FACS using the SH800S cell sorter (SONY) with a 100 μm sorting chip. We sorted pools of approximately 150,000 cells per line on the basis of NGFR expression.

**B16-F10 melanoma/T-cell co-cultures**. To study how antigen levels influence T-cell activation and cellular fitness in vitro we co-cultured our B16-F10 OVA-miSFIT cell lines with OT-I T-cells. Primary splenocytes were harvested from C57BL/6, OT-I mice and stimulated with SIINFEKL peptide (20 μg/mL Cambridge Peptides) and IL-2 (10 units/mL BioLegend) in RPMI-1640 (Gibco) supplemented with 10% FBS (Gibco), 1% P/S, (10,000 U/mL, Gibco), 10 mM HEPES (Life Technologies), 1 mM Sodium Pyruvate (Life Technologies), 50 μM 2-Mercaptoethanol (Gibco) and 1% MEM Non-Essential Amino Acids (Gibco). After 48 h, CD8$^+$ T-cells were isolated using the mouse CD8a$^+$ T-cell Isolation Kit (Miltenyi Biotec). For melanoma fitness experiments, 20,000 B16-F10 cells (approx. 50/50 mixture of B16-F10 OVA- cells and OVA$^+$, miSFIT cells) were seeded per well in a 96-well plate ($n = 3$ per cell line). After allowing B16-F10 cells to adhere for 3 h, OT-I T-cells were added to each well at different T-cell: B16-F10 ratios. Mixed cultures were incubated overnight and analysed by flow cytometry. Relative fitness was calculated by dividing the frequency of NGFR$^+$ cells in the +T-cell condition by the frequency of NGFR$^+$ cells in the no-T-cell condition. For T-cell activation experiments, stimulated T-cells were rested for 72 h prior to being co-cultured with individual OVA-miSFIT cell lines at an 8:1 T-cell to B16-F10 ratio ($n = 5$ per cell line). After 24 h, we analysed T-cells by flow-cytometry. Antibody clones and suppliers are listed in Supplementary Table 1.

**In vivo tumour growth assays**. For in vivo tumour growth assays, 150,000 B16-F10 OVA-miSFIT cells were intradermally injected into WT C57BL/6 recipient mice ($n = 6$ recipient mice per cell line) on day 0. We isolated OT-I T-cells and stimulated them for 48 h (see above and Fig. 4a) and intravenously injected 500,000 CD8$^+$, OT-I T-cells per recipient mouse on day 7. Following the T-cell infusion we measured tumours every second day using Digital Callipers (Fisher Scientific). The experimenter performing the measurement was blinded to tumour identity. We culled mice when tumours exceeded 95 mm$^2$ using approved methods. Mice that did not have detectable tumours by day 17 were excluded from the study.

For TIL analysis, tumours were injected as described above but OT-I T-cells were adoptively transferred on day 8 to reduce the likelihood of complete tumour clearance. On day 13 all mice were culled and spleens and tumours were harvested by dissection. Spleens were processed as described above and tumours were dissociated using the Tumour Dissociation Kit, mouse (Miltenyi Biotec). Cells were washed, blocked with TruStain fcX (Biolegend) and stained with antibodies as listed in Supplementary Table 1. Animal experiments were conducted under a project licence approved by an internal Oxford review board and the UK home office and were carried out in compliance with relevant regulations for animal testing and research.

**CRISPR/Cas9 mediated endogenous BRCA1 tuning**. A sgRNA targeting the spacer sequence 5′-AAGAGTGAGAGGAGCTCCCA-3′ in the BRCA1 3′UTR was cloned into the pSpCas9(BB)-2A-GFP expression vector (PX458, a gift from Feng Zhang, Addgene plasmid #48138). ssODN HDR donors designed to contain each miSFIT variant and a NheI recognition site flanked on either side by 45nt homology arms were synthesised by Integrated DNA technologies. HDR donors were designed to destroy an endogenous SacI site within the target locus. All 18 HDR donors (15 miSFIT variants, 1× and 2× perfectly complementary sites and a

scrambled MRE) were re-suspended to 100 μM and pooled. We nucleofected 1 × 10$^5$ HEK-293T cells with the sgRNA expression plasmid (500 ng) and the pooled HDR donors (0.2 μL, 100 μM stock) using the Neon transfection system (ThermoFisher Scientific) in a 10 μL tip (1150 V, 20 ms, 2 pulses). We performed 9 nucleofections which were pooled in groups of three to produce biological triplicates.

Seventy-two hours post-nucleofection, we simultaneously harvested genomic DNA and mRNA using the All Prep DNA/RNA Mini kit (Qiagen). Contaminating gDNA was eliminated from isolated RNA using the Turbo DNA-free kit (ThermoFisher Scientific) and cDNA was generated using the QuantiTect Reverse Transcription kit (Qiagen) following the manufacturer's instructions.

To create amplicon libraries for high-throughput sequencing, the targeted locus in the BRCA1 3′UTR was PCR amplified using the primers BRCA1-Miseq-F and BRCA1-Miseq-R. We used Phusion High-Fidelity PCR Master Mix with GC Buffer (NEB) and the following cycling conditions: initial denaturation (98 °C for 30 s), 33 amplification cycles (98 °C for 10 s, 64 °C for 12 s, 72 °C for 12 s) and final extension (72 °C for 5 min). We used 50 ng gDNA and cDNA generated from 200 ng of RNA as input for these PCRs. The PCRs were purified using the MinElute PCR Purification Kit (Qiagen). To enrich for CRISPR/Cas9-edited gDNA/cDNA, we digested WT DNA from these PCR products with SacI-HF (NEB) and gel-purified the remaining CRISPR/Cas9-modified, undigested PCR product (QIAquick Gel Extraction Kit, Qiagen). Illumina sequencing adaptors and unique barcode combinations were appended to the HDR-enriched DNA by PCR. We used Phusion High-Fidelity PCR Master Mix with GC Buffer (NEB) and the following cycling conditions: initial denaturation (98 °C for 30 s), 12 amplification cycles (98 °C for 10 s, 62 °C for 10 s, 72 °C for 10 s) and final extension (72 °C for 5 min). We purified, quantified and sequencing the resulting amplicon libraries as described above.

**Code availability**. Custom R scripts used to analyse HTS data are available from the corresponding author upon reasonable request.

**Reporting summary**. Further information on experimental design is available in the Nature Research Reporting Summary linked to this article.

## Data availability

Raw HTS data (Fastq files) have been deposited into the Sequence Read Archive (SRA) at the European Nucleotide Archive (ENA). SRA accession: PRJNA516224. All other raw data that is not found in the supplementary information is available from the corresponding author upon reasonable request. Relevant plasmids described in this study are available from Addgene (http://www.addgene.org/Tudor_Fulga/).

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

## Acknowledgements

We thank Jamie Michaels and Noam Prywes for their comments on the manuscript. We thank Paul Sopp, Kevin Clark, Sally-Ann Clark and Craig Waugh from the WIMM Flow Cytometry Facility for providing training and technical support. We acknowledge Justin Deme and Kathryn Robson for their advice and technical assistance in performing polysome profiling experiments. We thank Natalie Beadle and Elena Zanchini for their assistance with cloning constructs used in this study. M.K.T. is supported by a Marie Skłodowska-Curie Individual Fellowship (European Research Council, Horizon 2020). Y.S.M. is funded by the Clarendon Scholarship, the WIMM prize studentship and the Christopher Welch Fellowship. H.M.S. was supported by The Royal Society of NZ, Catalyst Seeding grant. D.J.H.F.K. is funded by a CIHR Postdoctoral Fellowship. V.C. and M.B.B. are funded by the MRC, Cancer Research UK (CRUK Programme C399/A2291) and the Oxford National Institute for Health Research (NIHR). T.A.M. is funded by Medical Research Council (MRC, UK) Molecular Haematology Unit Grant MC_UU_12009/6. T.A.B. was supported by a Radcliffe Department of Medicine/MRC Scholars Programme Studentship. T.A.F. was supported by MRC (G0902418), BBSRC (BB/N006550/1) and Wellcome Trust ISSF (105605/Z/14/Z).

## Author contributions

Y.S.M. conceived the study. Y.S.M., T.A.F., and M.B.B. designed the experiments and analysed the results. Y.S.M. and M.B.B. executed most of the experiments. H.B., T.A.B., M.K.T., and V.A. helped with the experiments and provided technical expertise. U.G. performed I.V. injections. H.C.-Y., M.F., H.M.S. and T.A.M. provided guidance and expertise. D.J.H.F.K. provided guidance and assisted with data analysis. V.C. provided expertise and helped with the experimental design. Y.S.M. and T.A.F. wrote the manuscript. All other authors provided feedback on the manuscript.

## Additional information

**Competing interests:** Y.S.M., M.B.B., T.A.M., V.C., and T.A.F. have filed a patent relating to the technology presented in this manuscript. T.A.M. is one of the founding shareholders of Oxstem Oncology (OSO), a subsidiary company of OxStem Ltd. The remaining authors declare no competing interests.

