## [Peer Review File · Nature Communications]

Reviewers' Comments:

Reviewer #1:

Remarks to the Author:

In general, the authors have successfully addressed all of the major concerns. However, as discussed below, there are a small number of minor concerns remaining, which the authors should address prior to publication.

To validate that their mRNA/pDNA sequencing-based assay successfully predicts protein levels, the authors performed polysome profiling (fig 1f-g) and analyzed some of the variants with FACS (supp figure 3d). Based on the FACS data, it is clear that the sequencing-based assay is validated. However, I still do not fully understand the interpretation of the polysome profiling experiment. The authors stated that they did this to "rule out the possibility that MRE variants were being translationally repressed in a manner that was not predicted by [their] mRNA/pDNA sequencing approach." However, it would appear that there is translational repression for the different MREs. Taken alone, the polysome experiment suggests that measuring mRNA levels would not be an accurate predictor of protein levels, as it would be missing the magnitude of this translational repression. However, the FACS data (in the supplement) suggests that there is no translational repression, and, indeed, the mRNA/pDNA sequencing is a good predictor of protein levels. Taken together, perhaps the decrease in polysome/monosome ratios is a consequence of the mRNA degradation rather than active translational repression, and is making a minor contribution to overall regulation. However, and importantly, the polysome/monosome ratios are not a measure of total translational output, which brings into question why these data would be used to validate the mRNA/pDNA sequencing (other than verifying that there are no single MRE variants that have outlying effects on translational efficiency). My view is that the current FACS data is adequate and ideal to validate the mRNA/pDNA sequencing to protein levels; therefore, I recommend moving that to figure 1 and including the polysome data to show that there are no MREs that give vastly aberrant translational efficiencies.

Reviewer #2:

Remarks to the Author:

Michaels et al. develop an elegant high-throughput screening strategy to evaluate the activity of microRNA-response elements (MRE). Their goal is to use them to fine tune with exquisite precision the expression of transgenes using endogenous microRNAs.

In the current revised version of the manuscript, the authors have addressed all the main concerns that I raised in the first version of their work. In particular:

- 1: My concern was that the authors were using a reporter with a microRNA response element (MRE) to define the upper limit of Ova expression. I was concerned that this reporter could still provide some degree of regulation, no matter how weak, compared to a reporter without MRE. The authors have provided enough experimental evidence that the with the mutation 17T actually represents the upper limit of expression. Specifically, they have cloned two scrambled MRE reporters (which are expected to have not targeting) and demonstrated that they achieve the same expression level as M17T.

-2: I expressed the concern about the potential of miSFITs to act microRNA sponges. Besides presenting bibliographical support, the authors conducted an additional experiment where at miR-17 reporter is co-transfected with different miSFITs. The strength of regulation of these miSFITs does not correlate with the strength of the miR-17 reporter regulation, suggestion that the miSFITs do not interfere or saturate the endogenous miR-17 activity.

3.- Through the analysis of single nucleotide MRE variants, the author achieved a gap width between reporters of 1.3%. Despite the impressive density of reporters, that gap did not

represented an analog continuum of regulators. The authors have amended the title and the text to acknowledge the digital nature of the reporters and also re-analyze their sequencing data to include the information arising from the miSFITs with two modifications. That reduced the gap between reporters to 0.022%, thus achieving near-analog resolution.

In addition, the authors have clarified the number of variants expected in their libraries and have performed an additional experiment where they integrated into the genome using CRISPR/Cas9 and HDR. Specifically, they integrated miSFITs into the 3'UTR of BRCA1, demonstrating that miSFITs are functional even when expressed from endogenous loci.

Therefore, I consider that the manuscript should be accepted in the current format for publication and that is suitable for the audience of Nature Communications.

Reviewer #3:

Remarks to the Author:

The improvement of the dynamic range through the use of multiple miRNA binding provides a good indication that this technology may be useful for transgene regulation. The demonstration of the endogenous gene regulation also strengthens the appeal of this technology. The in vivo data, however, remains to be the weakest part of this manuscript. At this moment, it seems that the chosen model is not ideally suited for miSFIT technology. With that being said, my enthusiasm for this manuscript has improved after the revision and I recommend publication.

Referee #1 (Remarks to the Author):

In general, the authors have successfully addressed all of the major concerns. [...] My view is that the current FACS data is adequate and ideal to validate the mRNA/pDNA sequencing to protein levels; therefore, I recommend moving that to figure 1 and including the polysome data to show that there are no MREs that give vastly aberrant translational efficiencies.

We thank the reviewer for their final comments and for their endorsement of our manuscript. Following the reviewer suggestion, we have now moved the FACS data from Supplementary Figure 3 into Figure 1, which also includes the relevant polysome data.

Referee #2 (Remarks to the Author):

Michaels et al. develop an elegant high-throughput screening strategy to evaluate the activity of microRNA-response elements (MRE). Their goal is to use them to fine tune with exquisite precision the expression of transgenes using endogenous microRNAs. [...] I consider that the manuscript should be accepted in the current format for publication and that is suitable for the audience of Nature Communications.

We thank the reviewer for their valuable feedback throughout the review process and for their final comments endorsing our manuscript.

Referee #3 (Remarks to the Author):

The improvement of the dynamic range through the use of multiple miRNA binding provides a good indication that this technology may be useful for transgene regulation. The demonstration of the endogenous gene regulation also strengthens the appeal of this technology. The in vivo data, however, remains to be the weakest part of this manuscript. At this moment, it seems that the chosen model is not ideally suited for miSFIT technology. With that being said, my enthusiasm for this manuscript has improved after the revision and I recommend publication.

We thank the reviewer for their helpful suggestions during the review process and for recommending the publication of our revised manuscript.